# Calibration and Identifiability of a Coupled Solow–SIR Economic–Epidemiological Model under Sparse Annual Observations

**S.A. Yakunin, O.I. Krivorotko**

## Abstract

This paper studies calibration and practical identifiability of a coupled economic–epidemiological model under sparse annual observations. The model combines a controlled SIR epidemic block with a parsimonious Solow-type reduced-form economic equation and is designed for settings where both economic and morbidity proxies are observed only at annual frequency.

The main contribution is a reproducible "forward simulator + inverse calibration" pipeline. On the forward side, we use within-year sub-stepping to reconcile the fast epidemic dynamics with annual observations and to compute annual prevalence- or incidence-type summaries. On the inverse side, we estimate a low-dimensional parameter vector by minimizing a composite objective that combines a profiled Gaussian negative log-likelihood with temporal regularization of residual increments, which improves numerical stability under short time series.

We also discuss practical identifiability issues arising under annual sampling, including parameter correlations and scale ambiguities, and show how bounded parameter domains, scale fixing, and reduced-form control parametrization improve robustness of calibration. A regional annual case study demonstrates that the proposed pipeline can produce consistent simultaneous fits of log-income and an annual morbidity proxy within a single mechanistic framework.

The proposed model should be interpreted as a transparent reduced-form baseline for structured joint modeling and scenario analysis rather than as a tool for causal identification. This work was supported by the grant of the state program of the "Sirius" Federal Territory "Scientific and technological development of the "Sirius" Federal Territory" (Agreement No. 26-03 dated 07.07.2025).

## 1 Introduction

Epidemic shocks affect economic outcomes through multiple channels, including reductions in effective labor supply, voluntary behavioral responses, and policy interventions. A widely used parsimonious modeling strategy is to couple a mechanistic epidemic core (SIR-type dynamics) (Kermack & McKendrick, 1927) with a macro growth component (Solow-type dynamics) (Solow, 1956), and to calibrate the resulting coupled system to observed economic and epidemiological time series. This class of models supports scenario analysis and policy counterfactuals and provides a transparent baseline for subsequent extensions toward optimal control and game-theoretic formulations (Eichenbaum et al., 2021; Alvarez et al., 2021; Acemoglu et al., 2021).

A central practical difficulty is *identification under sparse observations.* In many regional settings, both income proxies and morbidity proxies are available only at an annual frequency, while the latent epidemic dynamics evolves on a much shorter time scale. This induces weak identifiability of latent states and strong parameter correlations, making calibration a nontrivial inverse problem (Kaipio & Somersalo, 2005; Tarantola, 2005). Accordingly, the focus of this work is methodological: we develop a reproducible and numerically stable "forward simulator + inverse calibration" pipeline that remains well-posed under short annual series.

On the forward side, we implement a coupled SIR–growth simulator with sub-stepping within each year to bridge the scale mismatch between fast epidemic dynamics and annual observations. On the observation side, we support two regimes for the epidemic channel: a prevalence proxy (annual snapshot of $I(t)$) and an incidence proxy (annual integral of the infection flow), which is often more compatible with reported annual morbidity statistics. On the inverse side, we estimate a low-dimensional parameter vector by minimizing a composite objective based on a profiled Gaussian negative log-likelihood (NLL) of residuals (with the noise variance profiled out) and an additional temporal-smoothness penalty on residual increments, which regularizes the calibration when the time series is short. The optimization is performed as a black-box search using Optuna (TPE) (Akiba et al., 2019), consistent with robust parameter search in nonconvex settings (Bergstra & Bengio, 2012; Snoek et al., 2012). Importantly, the calibrated coupling should be interpreted as a structured reduced-form mechanism consistent with the observed annual trajectories, rather than as causal identification in the econometric sense. Under sparse annual observations, the goal is reproducible joint modeling and stable calibration, not causal attribution of economic changes to epidemic burden. To mitigate scale ambiguity between the initial state and the technology level, we fix the initial income scale by the first observation and optimize the technology level in log-scale.

**Contributions.** The main contributions are:

- a coupled SIR–growth forward simulator aligned with annual observations via within-year sub-stepping and explicit computation of annual incidence;
- a joint calibration objective that combines profiled Gaussian NLL with a residual-increment regularizer to stabilize estimation under short annual series;
- a reproducible black-box calibration procedure implemented with Optuna (TPE), including practically motivated parameter bounds and a scale-fixing strategy;
- a real-data regional case study illustrating the feasibility of simultaneous calibration to log-income and an annual morbidity proxy within a single coupled mechanistic framework.

**Control and future work.** To keep the inverse problem identifiable under annual data, we represent policy interventions in reduced form by a constant average intensity $u_{2,\text{const}}$ that modulates the contact rate. A full time-varying optimal control layer (e.g., Pontryagin maximum principle or dynamic programming) is deferred to future work (Pontryagin et al., 1962; Bellman, 1957), as are stochastic and mean-field extensions (Lasry & Lions, 2007; Huang et al., 2006).

**Paper organization.** Section 2 reviews related work. Section 3 describes the data and preprocessing. Section 4 introduces the coupled model. Section 5 specifies the observation model. Section 6 describes the numerical forward simulation and incidence computation. Section 7 formulates the inverse calibration problem and the optimization procedure. Section 8 reports the real-data experiment, and Sections 9–10 discuss limitations and conclude.

## 2 Related work

Our study lies at the intersection of (i) mechanistic epidemic modeling, (ii) epidemic–macroeconomic coupling, and (iii) statistical calibration of dynamical systems under partial and sparse observations.

**Epidemic dynamics and interpretable summary quantities.** The classical SIR framework was introduced by Kermack & McKendrick (1927) and remains a standard baseline for compartmental epidemic dynamics. Modern expositions emphasize analytical properties, threshold behavior, and the role of reproduction numbers; see, e.g., Hethcote (2000) and the monograph Anderson & May (1991). A systematic construction of next-generation matrices and associated threshold quantities for compartmental models is provided in Diekmann et al. (2010), which is useful when extending beyond the single-group SIR structure.

Table 1: Notation for annual observed series and the scales used in calibration.

| Symbol | Meaning | Scale used in calibration |
|---|---|---|
| $Y^{\mathrm{obs}}(t_k)$ | annual income/output proxy | $y_k^{\mathrm{obs}} = \log Y^{\mathrm{obs}}(t_k)$ |
| $i_k^{\mathrm{obs}}$ | annual morbidity proxy | $\tilde{i}_k^{\mathrm{obs}} = \log(\max\{i_k^{\mathrm{obs}}, \varepsilon\})$ |

**Epidemic–macroeconomic models.** During the COVID-19 period, SIR-based macroeconomic models became a prominent baseline for evaluating policy counterfactuals and welfare trade-offs. A representative reference is Eichenbaum et al. (2021), which couples epidemic dynamics with economic decisions and quantifies macroeconomic effects. Simplified planning formulations for lockdown-style interventions were proposed in Alvarez et al. (2021), while targeted interventions in multigroup SIR settings were studied in Acemoglu et al. (2021). Related economic mechanisms and externalities of distancing are discussed in Farboodi et al. (2021). In contrast to these works, our focus is not on a rich structural utility–policy layer but on a low-dimensional, identifiable calibration pipeline under annual observations, which can subsequently serve as a building block for control and game-theoretic extensions.

**Inverse problems, calibration, and identifiability under sparse observations.** Estimating parameters of nonlinear dynamical systems from noisy, partial observations is a classical inverse-problem task; see Kaipio & Somersalo (2005); Tarantola (2005) for general treatments. In the epidemic time-series context, mechanistic models are often embedded into statistical frameworks for partially observed Markov processes (POMP), enabling likelihood-based and "plug-and-play" inference (Bretó et al., 2009; Ionides et al., 2006; He et al., 2010; King et al., 2016). Iterated filtering provides a practical approach for maximum-likelihood-type estimation in such settings (Ionides et al., 2011; 2015). A persistent challenge is (practical) identifiability, especially for short and/or low-frequency time series; profile likelihood methods are widely used to diagnose parameter non-identifiability and to construct uncertainty intervals (Raue et al., 2009; Kreutz et al., 2013), including Monte Carlo profile confidence intervals for dynamic systems (Ionides et al., 2017). Motivated by these considerations, we adopt a profiled likelihood objective (profiling out the residual variance) combined with a simple temporal regularizer on residual increments, and we solve the resulting nonconvex problem via black-box optimization.

**Black-box optimization for nonconvex calibration.** When the calibration objective is nonconvex and expensive to evaluate, black-box search methods are commonly used. Random search is a strong baseline for hyperparameter tuning (Bergstra & Bengio, 2012), while Bayesian optimization methods have been developed for more sample-efficient exploration (Snoek et al., 2012). In this work we rely on Optuna and its TPE sampler (Akiba et al., 2019) as a reproducible implementation framework for bounded, low-dimensional parameter search.

**Connection to control and mean-field formulations.** Finally, once a baseline coupled simulator is identified, it can be extended toward optimal control (Pontryagin principle; dynamic programming) (Pontryagin et al., 1962; Bellman, 1957) and toward mean-field game formulations for large-population interacting systems (Huang et al., 2006; Lasry & Lions, 2007; Carmona & Delarue, 2018). These directions are outside the scope of the present calibration-focused study but motivate the emphasis on an interpretable and reproducible forward–inverse pipeline.

## 3 DATA AND PREPROCESSING

We consider a regional case study with annual observations at times $t_k = k$, $k = 0, \ldots, T-1$, after aligning calendar years. The dataset consists of (i) an economic time series $Y^{\mathrm{obs}}(t_k) > 0$ (an income/output proxy) and (ii) an annual morbidity proxy $i_k^{\mathrm{obs}}$.

**Economic series.** To reduce scale effects and to linearize multiplicative deviations, we use log-income observations

$$y_k^{\mathrm{obs}} = \log\big(Y^{\mathrm{obs}}(t_k)\big). \tag{1}$$

In the reported experiments, the initial condition of the economic state is fixed by the first observation,

$$Y(0) = Y^{\mathrm{obs}}(t_0), \tag{2}$$

which mitigates scale ambiguity between the initial state and the technology level in the Solow-type growth equation.

**Morbidity proxy and annual alignment.** The epidemiological channel is provided as an annual morbidity proxy $i_k^{\mathrm{obs}}$. Depending on reporting conventions, we interpret it either as a prevalence proxy (an annual snapshot of $I(t_k)$) or as an incidence proxy (annual cumulative infections over $[t_k, t_{k+1}]$), consistent with the observation regimes in Section 5. In annual reporting, an incidence-type proxy is often the most natural summary statistic.

**Transforms for the epidemic channel.** Annual morbidity values can be small in magnitude and may contain zeros due to reporting thresholds. To improve numerical stability and to balance scales between channels, we compare the epidemic proxy in a log-stabilized scale:

$$\tilde{i}_k^{\mathrm{obs}} = \log\big(\max\{i_k^{\mathrm{obs}}, \varepsilon\}\big), \tag{3}$$

with a fixed $\varepsilon > 0$ to avoid $\log(0)$. The model output $\tilde{i}_k(\theta)$ is computed from the simulated incidence or prevalence and transformed in the same way.

**Missing values and cleaning.** If missing values occur in either channel, we restrict the calibration window to years where both $y_k^{\mathrm{obs}}$ and $i_k^{\mathrm{obs}}$ are available. No additional smoothing is applied; stability is provided by the residual-increment penalty in the calibration objective.

**Descriptive summary.** We visualize the aligned observed series and the calibrated trajectories in Figure 1. Basic summary statistics for both channels are reported together with the main fit results in Section 8.

## 4 MODEL

We formulate a coupled economic–epidemiological dynamical system on a finite time horizon $t \in [0, T]$, measured in *years*. The state vector is

$$x(t) = (S(t), I(t), R(t), Y(t)) \in \mathbb{R}^4,$$

where $S, I, R$ are population shares (dimensionless) and $Y(t) > 0$ is an annual-scale income/output proxy (in the units of the data). The policy/intervention enters through a control $u_2(t) \in [0, 1]$ that modulates effective contacts.

### 4.1 EPIDEMIOLOGICAL BLOCK (CONTROLLED SIR)

We use a normalized SIR model (Kermack & McKendrick, 1927) with control-dependent contact rate:

$$\dot{S}(t) = -\beta(t)\, S(t)\, I(t), \tag{4}$$

$$\dot{I}(t) = \beta(t)\, S(t)\, I(t) - \gamma\, I(t), \tag{5}$$

$$\dot{R}(t) = \gamma\, I(t), \tag{6}$$

where $\gamma > 0$ is the removal rate (units: 1/year) and

$$\beta(t) = \beta_0\big(1 - u_2(t)\big), \qquad \beta_0 > 0, \quad u_2(t) \in [0, 1]. \tag{7}$$

Initial conditions are

$$S(0) = 1 - I_0, \qquad I(0) = I_0, \qquad R(0) = 0, \qquad I_0 \in (0, 1). \tag{8}$$

**Admissible controls.** We allow measurable bounded controls

$$u_2 \in \mathcal{U} := \{u \in L^\infty([0,T]) : 0 \le u(t) \le 1 \text{ a.e.}\},$$

and in the identification experiments we use the reduced-form parametrization

$$u_2(t) \equiv u_{2,\mathrm{const}} \in [0,1]. \tag{9}$$

**Invariant set and feasibility.** Define the simplex

$$\Delta := \{(S,I,R) \in \mathbb{R}^3 : S \ge 0, \ I \ge 0, \ R \ge 0, \ S + I + R = 1\}.$$

Summing (4)–(6) yields $\frac{d}{dt}(S + I + R) = 0$, hence $S + I + R \equiv 1$ if (8) holds. Moreover, the vector field points inward on the boundary of $\Delta$ (e.g., if $I = 0$ then $\dot{I} = 0$, if $S = 0$ then $\dot{S} = 0$, etc.), so $(S(t), I(t), R(t)) \in \Delta$ for all $t \in [0,T]$.

**Reproduction numbers.** Under (7), the basic reproduction number in the constant-control regime (9) is

$$R_0 = \frac{\beta_0(1 - u_{2,\mathrm{const}})}{\gamma}, \tag{10}$$

and the instantaneous effective reproduction number is

$$R_e(t) = \frac{\beta(t)}{\gamma} S(t) = \frac{\beta_0(1 - u_2(t))}{\gamma} S(t). \tag{11}$$

## 4.2 Economic block (Solow-type reduced form with epidemic burden)

Let $Y(t) > 0$ denote an income/output proxy (e.g., per-capita income). We use a parsimonious Solow-type reduced-form dynamics with an exogenous technology trend $A(t)$ and an epidemic burden factor:

$$\dot{Y}(t) = A(t) Y(t)^\alpha (1 - I(t))^{1-\alpha} - \mu Y(t), \qquad \alpha \in (0,1), \ \mu > 0, \tag{12}$$

where the technology level follows an exponential trend

$$A(t) = A_0 e^{gt}, \qquad A_0 > 0, \quad g \in \mathbb{R}. \tag{13}$$

The multiplicative term $(1 - I(t))^{1-\alpha}$ represents a reduced-form attenuation of effective production associated with higher infection prevalence. The specification (12) is deliberately low-dimensional to improve practical identifiability under annual observations and should be interpreted as a parsimonious phenomenological coupling rather than a causal structural economic model.

**Nonnegativity of the economic state.** On $\Delta$ we have $A(t) > 0$ and $(1 - I(t))^{1-\alpha} \ge 0$. For $Y > 0$, the right-hand side of (12) is continuous and locally Lipschitz in $Y$. Moreover, $Y(t) \equiv 0$ is an equilibrium of (12); hence $Y(t) \ge 0$ is forward invariant. In all experiments we enforce $Y(0) = Y^{\mathrm{obs}}(t_0) > 0$ and use numerical safeguards (Section 6) to prevent $Y$ from approaching 0 in discrete time.

## 4.3 Coupled system and well-posedness

Collecting (4)–(6) and (12), we obtain the coupled ODE

$$\dot{x}(t) = f(t, x(t); \theta, u_2(t)), \qquad x(0) = x_0(\theta), \tag{14}$$

with parameter vector

$$\theta = (\mu, \log A_0, g, \beta_0, \gamma, I_0, u_{2,\mathrm{const}}), \tag{15}$$

and constraints $\mu > 0$, $\beta_0 > 0$, $\gamma > 0$, $I_0 \in (0,1)$, $u_{2,\mathrm{const}} \in [0,1]$. For any admissible control $u_2 \in \mathcal{U}$, the vector field is measurable in $t$ and locally Lipschitz in $x$ on $\Delta \times (0,\infty)$; hence the coupled system admits a unique Carathéodory solution on $[0,T]$. Furthermore, the feasible set

$$\Omega := \Delta \times (0,\infty)$$

is forward invariant: $(S(t), I(t), R(t)) \in \Delta$ and $Y(t) > 0$ for all $t \in [0,T]$ whenever $x(0) \in \Omega$.

### 4.4 MODEL-IMPLIED ANNUAL SUMMARIES (PREVALENCE VS. INCIDENCE)

Let annual observation times be $t_k = k$, $k = 0, \ldots, T - 1$ (after calendar alignment). We compare log-income

$$y_k(\theta) = \log Y(t_k; \theta), \tag{16}$$

with observations $y_k^{\text{obs}} = \log(Y^{\text{obs}}(t_k))$.

For the epidemic channel we define the model-implied annual proxy $i_k(\theta)$ in one of two regimes:

- **Prevalence regime:**  $i_k(\theta) = I(t_k; \theta)$,  $k = 0, \ldots, T - 1$.
- **Incidence regime:**

$$i_k(\theta) = \int_{t_k}^{t_{k+1}} \beta(t)\, S(t; \theta)\, I(t; \theta)\, dt, \qquad k = 0, \ldots, T - 2, \tag{17}$$

  which corresponds to the cumulative share of new infections over year $k$ in the SIR block.

When morbidity values are small, we compare in a stabilized scale

$$\tilde{i}_k = \log(\max\{i_k, \varepsilon\}), \qquad \varepsilon > 0, \tag{18}$$

applied consistently to both observed and simulated epidemic proxies.

**Scale fixing for the economic state.**  To reduce scale ambiguity between the technology level and the initial condition under annual sampling, we fix

$$Y(0) = Y^{\text{obs}}(t_0), \tag{19}$$

and optimize $A_0$ in log-scale via $\log A_0$.

**Practical identifiability remark.**  Under the constant-control parametrization (9), the epidemic dynamics depends on $\beta_0$ and $u_{2,\text{const}}$ primarily through the product $\beta_0(1 - u_{2,\text{const}})$ (cf. (7)–(10)), so separating these parameters from annual data alone is typically ill-conditioned. This motivates bounded domains, reporting derived effective quantities (e.g., $R_0$), and the reduced-form control used in the calibration pipeline.

## 5 OBSERVATION MODEL

We assume that observations are available at annual times $t_k = k$, $k = 0, \ldots, T - 1$ after calendar alignment. Let $x(t) = (S(t), I(t), R(t), Y(t))$ be the solution of the coupled system for a parameter vector $\theta$. The observation model specifies (i) which functions of the latent state are compared to data and (ii) the statistical structure of measurement errors. Throughout, we keep the likelihood intentionally low-dimensional to avoid over-parameterization under short annual series.

### 5.1 ECONOMIC CHANNEL: LOG-INCOME OBSERVATIONS

The economic data are positive annual observations $Y^{\text{obs}}(t_k) > 0$. We compare them in log-scale,

$$y_k^{\text{obs}} = \log\big(Y^{\text{obs}}(t_k)\big), \qquad y_k(\theta) = \log\big(Y(t_k; \theta)\big), \tag{20}$$

and define residuals

$$r_k^Y(\theta) = y_k^{\text{obs}} - y_k(\theta), \qquad k = 0, \ldots, T - 1. \tag{21}$$

This corresponds to a multiplicative-noise interpretation in the original (non-log) scale.

## 5.2 Epidemic channel: prevalence vs. incidence proxies

Let $i_k^{\mathrm{obs}}$ denote an annual morbidity proxy. We map the latent SIR state to a model-implied annual proxy $i_k(\theta)$ according to Section 4.4:

$$i_k(\theta) = \begin{cases} I(t_k; \theta), & \text{prevalence regime,} \quad k = 0, \ldots, T-1, \\ \int_{t_k}^{t_{k+1}} \beta(t) S(t; \theta) I(t; \theta)\, dt, & \text{incidence regime,} \quad k = 0, \ldots, T-2. \end{cases} \tag{22}$$

## 5.3 Stabilized comparison scale

Annual morbidity proxies can be small in magnitude and may contain zeros due to reporting thresholds or aggregation. To obtain a numerically stable and scale-balanced comparison, we use a log-stabilized transform

$$\tilde{i}_k^{\mathrm{obs}} = \log\big(\max\{i_k^{\mathrm{obs}}, \varepsilon\}\big), \qquad \tilde{i}_k(\theta) = \log\big(\max\{i_k(\theta), \varepsilon\}\big), \tag{23}$$

with a fixed $\varepsilon > 0$. Residuals for the epidemic channel are then defined as

$$r_k^I(\theta) = \tilde{i}_k^{\mathrm{obs}} - \tilde{i}_k(\theta). \tag{24}$$

## 5.4 Noise model and profiled Gaussian likelihood

We adopt an additive Gaussian noise model for residuals in each channel:

$$r_k^Y(\theta) \sim \mathcal{N}(0, \sigma_Y^2), \qquad r_k^I(\theta) \sim \mathcal{N}(0, \sigma_I^2), \tag{25}$$

independently across $k$ within each channel.[1] Rather than estimating $\sigma_Y^2$ and $\sigma_I^2$ explicitly (which is unstable for short series), we use the profiled likelihood: for any residual vector $r = (r_1, \ldots, r_n)$, the maximum-likelihood variance estimate is

$$\hat{\sigma}^2(r) = \frac{1}{n} \sum_{k=1}^{n} r_k^2, \tag{26}$$

and the corresponding profiled negative log-likelihood is

$$\mathrm{NLL}(r) = \frac{n}{2}\Big(1 + \log\big(2\pi\hat{\sigma}^2(r)\big)\Big). \tag{27}$$

This is the likelihood term used in the composite calibration objective.

## 5.5 Channel balancing

Because the economic and epidemic channels can have different scales even after transforms, we combine their losses via weights $W_Y, W_I > 0$ (see Section 7):

$$J(\theta) = W_Y L_Y(\theta) + W_I L_I(\theta), \tag{28}$$

where $L_Y$ and $L_I$ are constructed from (27) plus temporal regularization on residual increments.

**Remark (identifiability under annual sampling).** The observation frequency is substantially lower than the intrinsic time scale of the SIR dynamics. Consequently, the prevalence regime is often weakly informative unless prevalence varies slowly, whereas the incidence regime aggregates the fast dynamics over the year and can be more stable for annual morbidity series. In both regimes, separating $\beta_0$ from $u_{2,\mathrm{const}}$ is typically ill-conditioned without additional information, since they enter predominantly through $\beta_0(1 - u_{2,\mathrm{const}})$.

## 6 Numerical methods

This section describes the forward solver used to evaluate the annual summaries (20) and (22) and the calibration objective. Because the epidemic dynamics typically evolves on a time scale shorter than one year, while observations are annual, we integrate the coupled system using within-year sub-stepping.

---

[1]This independence assumption is a working approximation under annual data. Temporal dependence is partially addressed through the residual-increment regularization introduced in Section 7.

## 6.1 TIME DISCRETIZATION

We discretize each year $[k, k + 1]$ into $n_{\text{sub}} \in \mathbb{N}$ uniform sub-steps with step size

$$\Delta t = \frac{1}{n_{\text{sub}}}. \tag{29}$$

Define grid points

$$t_{k,\ell} = k + \ell \Delta t, \qquad \ell = 0, 1, \ldots, n_{\text{sub}}.$$

We denote the numerical states by

$$(S_{k,\ell}, I_{k,\ell}, R_{k,\ell}, Y_{k,\ell}) \approx (S(t_{k,\ell}), I(t_{k,\ell}), R(t_{k,\ell}), Y(t_{k,\ell})).$$

## 6.2 EXPLICIT EULER UPDATES FOR THE COUPLED DYNAMICS

On each sub-step we apply an explicit Euler update to the SIR block and the economic state. Let

$$\beta_{k,\ell} = \beta_0 \big(1 - u_2(t_{k,\ell})\big) \tag{30}$$

and

$$A_{k,\ell} = \exp(\log A_0) \, e^{g t_{k,\ell}}. \tag{31}$$

Then the explicit Euler updates read:

$$\text{flow}_{k,\ell} := \beta_{k,\ell} \, S_{k,\ell} \, I_{k,\ell}, \tag{32}$$

$$S_{k,\ell+1} = S_{k,\ell} - \Delta t \, \text{flow}_{k,\ell}, \tag{33}$$

$$I_{k,\ell+1} = I_{k,\ell} + \Delta t \big(\text{flow}_{k,\ell} - \gamma I_{k,\ell}\big), \tag{34}$$

$$R_{k,\ell+1} = R_{k,\ell} + \Delta t \, \gamma I_{k,\ell}, \tag{35}$$

$$Y_{k,\ell+1} = Y_{k,\ell} + \Delta t \Big(A_{k,\ell} \, Y_{k,\ell}^{\alpha} \, (1 - I_{k,\ell})^{1-\alpha} - \mu Y_{k,\ell}\Big). \tag{36}$$

In the constant-control regime $u_2(t) \equiv u_{2,\text{const}}$, we have $\beta_{k,\ell} \equiv \beta_0(1 - u_{2,\text{const}})$ for all grid points.

## 6.3 FEASIBILITY SAFEGUARDS FOR SIR AND POSITIVITY OF $Y$

Due to discretization error, explicit Euler may produce small infeasibilities (e.g., negative components) if $\Delta t$ is not sufficiently small. To preserve the SIR simplex and numerical stability, after each step we apply a simple projection onto

$$\Delta := \{(S, I, R) \in \mathbb{R}^3 : S \geq 0, \ I \geq 0, \ R \geq 0, \ S + I + R = 1\}.$$

Concretely, letting $(\bar{S}, \bar{I}, \bar{R}) = (S_{k,\ell+1}, I_{k,\ell+1}, R_{k,\ell+1})$, we set

$$S_{k,\ell+1} \leftarrow \min\{1, \max\{0, \bar{S}\}\}, \qquad I_{k,\ell+1} \leftarrow \min\{1, \max\{0, \bar{I}\}\}, \qquad R_{k,\ell+1} \leftarrow \min\{1, \max\{0, \bar{R}\}\}, \tag{37}$$

$$(S_{k,\ell+1}, I_{k,\ell+1}, R_{k,\ell+1}) \leftarrow \frac{(S_{k,\ell+1}, I_{k,\ell+1}, R_{k,\ell+1})}{S_{k,\ell+1} + I_{k,\ell+1} + R_{k,\ell+1}} \quad \text{whenever } S_{k,\ell+1} + I_{k,\ell+1} + R_{k,\ell+1} > 0. \tag{38}$$

(If the sum is zero after clipping, we set $(S_{k,\ell+1}, I_{k,\ell+1}, R_{k,\ell+1}) = (1, 0, 0)$.)

For the economic state we enforce a small positive floor

$$Y_{k,\ell+1} \leftarrow \max\{Y_{k,\ell+1}, \varepsilon_Y\}, \qquad \varepsilon_Y > 0, \tag{39}$$

to keep the log-transform well-defined and to avoid numerical issues when evaluating $Y^{\alpha}$.

## 6.4 DISCRETE OBSERVATION EXTRACTION AND INCIDENCE APPROXIMATION

We initialize $(S_{0,0}, I_{0,0}, R_{0,0}, Y_{0,0})$ from (8) and (19), and for each year $k$ we propagate (33)–(36) for $\ell = 0, \ldots, n_{\text{sub}} - 1$. Year-boundary states satisfy the recursion

$$(S_{k+1,0}, I_{k+1,0}, R_{k+1,0}, Y_{k+1,0}) := (S_{k,n_{\text{sub}}}, I_{k,n_{\text{sub}}}, R_{k,n_{\text{sub}}}, Y_{k,n_{\text{sub}}}).$$

We extract annual values at boundaries

$$Y_k := Y_{k,0}, \qquad I_k := I_{k,0}, \qquad k = 0, \ldots, T - 1, \tag{40}$$

and define model-implied log-income $y_k = \log Y_k$.

**Incidence approximation.** In the incidence regime, the annual infection flow over year $k$ is approximated by a Riemann sum consistent with (17):

$$i_k := \sum_{\ell=0}^{n_{\text{sub}}-1} \text{flow}_{k,\ell}\, \Delta t = \sum_{\ell=0}^{n_{\text{sub}}-1} \beta_{k,\ell}\, S_{k,\ell}\, I_{k,\ell}\, \Delta t, \qquad k = 0, \ldots, T-2. \tag{41}$$

This discrete incidence is then transformed via $\tilde{i}_k = \log(\max\{i_k, \varepsilon\})$ as in (23).

## 6.5 Numerical accuracy considerations

The explicit Euler scheme has first-order global accuracy $O(\Delta t)$ for sufficiently smooth solutions. In our setting, sub-stepping serves two purposes: (i) stabilizing integration of the (potentially fast) SIR dynamics and (ii) approximating the annual incidence functional reliably. In practice, $n_{\text{sub}}$ is chosen so that annual summaries $(y_k, i_k)$ are insensitive to further refinement.

**Algorithmic summary.** Given $\theta$ and a control $u_2$ (constant in the reported experiments), the solver iterates (33)–(36) over $k = 0, \ldots, T-2$ and $\ell = 0, \ldots, n_{\text{sub}}-1$, applies the safeguards (37)–(39), accumulates incidence via (41), and returns annual summaries $(y_k, i_k)$ for evaluation of the calibration objective.

# 7 Inverse problem and optimization

We formulate joint calibration as an inverse problem: find parameters $\theta$ such that the model-implied annual summaries match the observed economic and morbidity series under short annual samples.

## 7.1 Parameter vector and admissible set

We estimate the low-dimensional parameter vector

$$\theta = (\mu, \log A_0, g, \beta_0, \gamma, I_0, u_{2,\text{const}}) \in \Theta, \tag{42}$$

with constraints

$$\mu > 0, \quad \beta_0 > 0, \quad \gamma > 0, \quad I_0 \in (0,1), \quad u_{2,\text{const}} \in [0,1],$$

and we optimize $A_0$ in log-scale to enforce positivity. The economic initial condition is fixed by the first observation,

$$Y(0) = Y^{\text{obs}}(t_0), \tag{43}$$

to mitigate scale ambiguity between $A_0$ and $Y(0)$ under annual sampling.

**Practical identifiability remark.** Under the constant-control parametrization, the SIR dynamics depends on $(\beta_0, u_{2,\text{const}})$ primarily through $\beta_0(1 - u_{2,\text{const}})$. As a result, separating $\beta_0$ from $u_{2,\text{const}}$ from annual data alone is typically ill-conditioned. We therefore employ bounded domains and report $R_0 = \beta_0(1 - u_{2,\text{const}})/\gamma$ as an interpretable summary.

## 7.2 Residuals and channel-wise profiled likelihood

Let $\mathcal{K}_Y = \{0, \ldots, T-1\}$ denote the economic years used in calibration. For the epidemic channel, we set $\mathcal{K}_I = \{0, \ldots, T-1\}$ in the prevalence regime and $\mathcal{K}_I = \{0, \ldots, T-2\}$ in the incidence regime. Given $\theta$, the solver (Section 6) returns annual summaries $(y_k(\theta), i_k(\theta))$ and we define residuals

$$r_k^Y(\theta) = y_k^{\text{obs}} - y_k(\theta), \qquad k \in \mathcal{K}_Y, \qquad\qquad r_k^I(\theta) = \tilde{i}_k^{\text{obs}} - \tilde{i}_k(\theta), \qquad k \in \mathcal{K}_I, \tag{44}$$

where $\tilde{i}$ is the stabilized transform (23).

For any residual vector $r = (r_1, \ldots, r_n)$, we use the profiled Gaussian negative log-likelihood (variance profiled out):

$$\text{NLL}(r) = \frac{n}{2}\Big(1 + \log\big(2\pi\hat{\sigma}^2(r)\big)\Big), \qquad \hat{\sigma}^2(r) = \frac{1}{n}\sum_{j=1}^{n} r_j^2. \tag{45}$$

### 7.3 Temporal regularization of residual increments

Annual series are short and may lead to overfitting to local fluctuations. To stabilize calibration, we add a penalty on residual increments:

$$\mathcal{R}(r) = \frac{1}{n-1} \sum_{j=1}^{n-1} \left( r_{j+1} - r_j \right)^2. \tag{46}$$

### 7.4 Composite objective

Channel-wise losses are

$$L_Y(\theta) = \text{NLL}\left( r^Y(\theta) \right) + \lambda_Y \, \mathcal{R}\left( r^Y(\theta) \right), \qquad L_I(\theta) = \text{NLL}\left( r^I(\theta) \right) + \lambda_I \, \mathcal{R}\left( r^I(\theta) \right), \tag{47}$$

and the total calibration objective is

$$J(\theta) = W_Y \, L_Y(\theta) + W_I \, L_I(\theta), \qquad \theta^\star \in \arg\min_{\theta \in \Theta} J(\theta), \tag{48}$$

with weights $W_Y, W_I > 0$ controlling channel balance and $\lambda_Y, \lambda_I \geq 0$ controlling temporal regularization strength. In the reported experiments we use equal channel weights $W_Y = W_I = 1$.

### 7.5 Black-box optimization with Optuna (TPE)

We minimize (48) by black-box optimization using Optuna with the TPE sampler (Akiba et al., 2019). Each objective evaluation consists of: (i) running the forward solver at $\theta$, (ii) computing annual summaries in the chosen regime, and (iii) forming residuals (44) and the loss (48).

**Search domain.** A typical bounded domain used in our experiments is:

$$\mu \in [10^{-4}, 2], \quad \log A_0 \in [-10, 5], \quad g \in [-0.1, 0.1], \quad \beta_0, \gamma \in [10^{-3}, 50], \quad I_0 \in [10^{-8}, 2 \cdot 10^{-2}], \quad u_{2,\text{const}} \in [0, 0.8].$$

We use log-uniform sampling for strictly positive parameters (e.g., $\mu, \beta_0, \gamma$) and uniform sampling for bounded parameters (e.g., $u_{2,\text{const}}$ and $g$).

**Reproducibility.** We fix the sampler seed and report the best trial parameters $\theta^\star$ together with the associated fit metrics.

## 8 Experiments and results

We report a regional annual case study and evaluate the proposed calibration pipeline on two coupled channels: log-income and an annual morbidity proxy. All experiments use the forward solver of Section 6 and the calibration objective of Section 7. Unless stated otherwise, we fix the production exponent $\alpha \in (0, 1)$ and set the initial economic scale via $Y(0) = Y^{\text{obs}}(t_0)$.

### 8.1 Experimental setup

We run calibration under the chosen epidemic observation regime (prevalence or incidence; Section 5). In the incidence regime, the model-implied annual proxy is the cumulative infection flow (17), approximated by (41).

### 8.2 Fit quality metrics

To quantify goodness-of-fit we report root mean squared error (RMSE) in each channel:

$$\text{RMSE}_Y = \sqrt{\frac{1}{|\mathcal{K}_Y|} \sum_{k \in \mathcal{K}_Y} \left( r_k^Y(\theta^\star) \right)^2}, \qquad \text{RMSE}_I = \sqrt{\frac{1}{|\mathcal{K}_I|} \sum_{k \in \mathcal{K}_I} \left( r_k^I(\theta^\star) \right)^2}, \tag{49}$$

with $\mathcal{K}_Y$ and $\mathcal{K}_I$ as defined in Section 7.

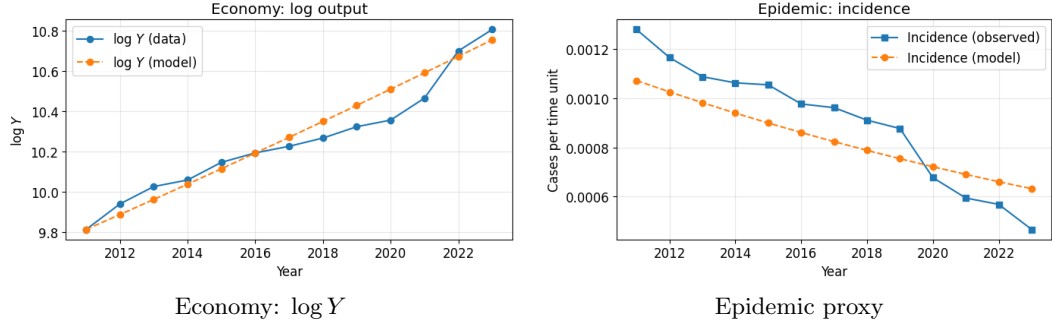

Figure 1: Observed vs. simulated trajectories for log-income (left) and the annual epidemic proxy (right) under the calibrated parameters.

Table 2: Best parameters returned by the Optuna optimization (reported run).

| Parameter | Value | Notes |
|---|---|---|
| $\mu$ | $3.0283 \times 10^{-4}$ | output decay / depreciation |
| $\log A_0$ | 3.98214 | optimized in log-scale |
| $A_0 = \exp(\log A_0)$ | 53.6317 | technology level |
| $g$ | 0.0618337 | technology growth rate |
| $\beta_0$ | 0.113108 | baseline contact rate |
| $\gamma$ | 0.128245 | removal rate |
| $I_0$ | 0.0127936 | initial infected share |
| $u_{2,\text{const}}$ | 0.233608 | constant intervention intensity |
| $\beta_0(1 - u_{2,\text{const}})$ | 0.0866851 | effective contact rate |
| $R_0 = \beta_0(1 - u_{2,\text{const}})/\gamma$ | 0.675933 | implied reproduction number |

## 8.3 MAIN CALIBRATION RESULT

Figure 1 compares observed and model-implied trajectories for both channels under the best-fit parameters. Table 2 reports the corresponding parameter estimates and the implied reproduction number $R_0$.

**Summary.** The experiment illustrates the feasibility of simultaneous calibration to annual log-income and an annual morbidity proxy within a single coupled mechanistic model, providing an interpretable baseline for subsequent extensions (time-varying control, stochasticity, and mean-field formulations).

**Identifiability note.** The reported parameter values should be interpreted with care: under annual sampling, several parameters are only practically identifiable up to correlated combinations. In particular, the epidemic channel is more informative about effective quantities such as $\beta_0(1 - u_{2,\text{const}})$ and $R_0$ than about $\beta_0$ and $u_{2,\text{const}}$ separately. The present experiment is therefore intended to demonstrate calibration feasibility and joint fit consistency rather than full parameter recoverability.

## 9 DISCUSSION AND LIMITATIONS

We discuss what the calibrated coupled model can and cannot support under annual observations, and we highlight limitations that motivate subsequent methodological extensions.

### 9.1 SCOPE AND VALUE OF THE PROPOSED PIPELINE

The main outcome of this work is a reproducible identification pipeline for a coupled mechanistic model under sparse annual sampling. Practically, it provides:

- a stable forward simulator that bridges the time-scale mismatch via within-year sub-stepping and enables consistent computation of annual incidence proxies;

- a joint calibration objective that remains numerically stable under short time series by combining a profiled likelihood term (variance profiled out) with mild temporal regularization;

- interpretable parameter estimates and derived summaries (e.g., $R_0$) that enable scenario simulation and counterfactual analysis within the reduced-form setting.

As a result, the model serves as an interpretable baseline for structured joint modeling of economic and epidemiological annual trajectories and for producing coherent fits consistent with both observed data channels.

## 9.2 Interpretation and non-causal scope

A good simultaneous fit of the economic and epidemiological channels should not be interpreted as causal identification of the effect of epidemic burden on economic outcomes. Under sparse annual observations, the calibrated coupled system provides a structured reduced-form description of joint dynamics that is useful for scenario analysis and consistency checks, but it does not by itself establish causality in the econometric sense. In particular, correlated trends, omitted mechanisms, and measurement aggregation may all contribute to the observed comovement.

## 9.3 Identifiability under annual sampling

A fundamental limitation is weak practical identifiability of latent epidemic dynamics and transmission parameters under annual observations. Even with the incidence functional (17), different parameter combinations can produce similar annual aggregates, yielding strong correlations and flat directions in the objective landscape. In particular, under the constant-control specification, $\beta_0$ and $u_{2,\text{const}}$ enter predominantly through the product $\beta_0(1 - u_{2,\text{const}})$, so separating these parameters is ill-conditioned without additional information. This motivates reporting derived effective quantities (e.g., $R_0$), using bounded domains, and keeping the control layer low-dimensional in the baseline calibration. In the present baseline setting, practical identifiability should therefore be understood as partial rather than absolute: some effective combinations and derived summaries can be estimated more robustly than individual primitive parameters. This is precisely why we emphasize scale fixing, bounded domains, reduced-form control parametrization, and reporting of effective quantities such as $R_0$.

## 9.4 Simplifications in the economic block

The economic equation (12) is intentionally compact: it encodes decreasing returns via $Y^\alpha$, long-run trend via $A(t)$, and epidemic suppression via $(1 - I)^{1-\alpha}$. At the same time, it abstracts away mechanisms that may matter empirically, including (i) capital accumulation and investment/consumption decisions, (ii) sectoral heterogeneity and labor market frictions, and (iii) delayed effects and adjustment costs. Accordingly, the economic block should be viewed as a reduced-form baseline. A natural next step is to introduce an explicit capital state with investment dynamics (Solow with capital), or a small structural core, once data availability supports stronger identification.

## 9.5 Measurement and proxy limitations

Both channels rely on proxies measured at a coarse annual frequency. The morbidity proxy may be affected by reporting conventions, changes in testing intensity, and aggregation artifacts, while zeros may reflect reporting thresholds rather than true absence of cases. The stabilized transform (23) improves numerical robustness but does not replace a dedicated measurement model. Similarly, the log transform for income reduces scale imbalance and corresponds to multiplicative deviations, but it cannot fully account for structural breaks or revisions in regional statistics.

## 9.6 Numerical considerations

We employ an explicit Euler scheme with feasibility projections for transparency and control of invariants. While first-order discretization is sufficient for the annual horizon and for calibration purposes, the method introduces a discretization bias of order $O(\Delta t)$. In practice, this bias is controlled by choosing $n_{\mathrm{sub}}$ large enough so that annual summaries are insensitive to further refinement. Higher-order solvers (e.g., Runge–Kutta methods) can be substituted, but they do not address the dominant limitation in this setting: identifiability under sparse data.

## 9.7 Outlook: toward control, stochasticity, and mean-field formulations

The present pipeline is designed as a baseline for three extensions. First, replacing the constant intervention intensity by a time-varying control $u_2(t)$ enables explicit optimal control formulations, but requires either richer data or stronger regularization/priors to remain identifiable. Second, stochastic extensions (process noise and explicit measurement noise) align the model with partially observed Markov process formulations and enable uncertainty quantification. Third, mean-field extensions can incorporate heterogeneity and strategic interactions; a calibrated forward core is a prerequisite for such developments. Overall, the proposed pipeline provides a stable starting point for these directions while remaining transparent and reproducible under annual data.

## 10 Conclusion and future work

We presented a coupled economic–epidemiological model that combines a controlled SIR block with a parsimonious Solow-type reduced-form economic dynamics and addressed the practical problem of calibration under sparse annual observations. The main methodological contribution is a reproducible "forward simulator + inverse calibration" pipeline: a within-year sub-stepping solver aligned with annual observation regimes (prevalence and incidence proxies), together with a joint calibration objective based on a profiled Gaussian negative log-likelihood and temporal regularization of residual increments. In a real-data regional annual case study, the proposed pipeline demonstrates the feasibility of simultaneous calibration to the log-income series and an annual morbidity proxy, providing an interpretable reduced-form baseline for structured joint modeling and future methodological extensions. The calibrated model should be interpreted as a tool for joint description and scenario analysis rather than causal identification.

**Future work.** The proposed baseline enables several direct extensions:

- **Time-varying interventions and optimal control:** replace the constant intervention intensity by a time-dependent control $u_2(t)$ and formulate an optimal control problem balancing economic and epidemiological objectives, supported by stronger priors/regularization or richer data.

- **Stochastic formulations and uncertainty quantification:** introduce process noise and an explicit measurement model, enabling likelihood-based or filtering-based inference and principled confidence intervals for parameters and trajectories.

- **Identifiability diagnostics:** augment point estimation with profile likelihood or Monte Carlo profile confidence intervals to quantify practical non-identifiability and to report uncertainty intervals for effective quantities such as $R_0$.

- **Richer economic core:** extend the reduced-form growth equation by introducing a latent capital state and explicit investment/consumption controls, improving the mapping between epidemic burden, labor supply, and long-run growth.

- **Mean-field and multi-agent extensions:** use the calibrated coupled simulator as a core component in mean-field control/game formulations with heterogeneous agents and strategic interactions.

## A    Implementation and numerical details

### A.1    Feasibility projections for the SIR simplex

To preserve the invariant set

$$\Delta = \{(S, I, R) \in \mathbb{R}^3 : \ S \geq 0, \ I \geq 0, \ R \geq 0, \ S + I + R = 1\},$$

we apply, after each Euler step, a simple projection consisting of:

1. componentwise clipping to $[0, 1]$,
2. renormalization $(S, I, R) \leftarrow (S, I, R)/(S + I + R)$ whenever $S + I + R > 0$.

This projection is a numerical safeguard; with sufficiently small $\Delta t$ the raw Euler update remains close to $\Delta$.

### A.2    Incidence computation and discretization error

In the incidence regime, the annual morbidity proxy is

$$i_k(\theta) = \int_{t_k}^{t_{k+1}} \beta(t) \, S(t) \, I(t) \, dt,$$

and the simulator uses the Riemann-sum approximation

$$i_k(\theta) \approx \sum_{\ell=0}^{n_{\text{sub}}-1} \beta_{k,\ell} S_{k,\ell} I_{k,\ell} \Delta t.$$

Because explicit Euler is first-order accurate, both the state trajectory and the incidence approximation have global discretization error of order $O(\Delta t)$.

### A.3    Log-stabilized transform for small morbidity values

To avoid numerical issues with $\log(0)$ and to stabilize comparisons when annual morbidity values are small, we use

$$\tilde{i} = \log(\max\{i, \varepsilon\})$$

with a fixed $\varepsilon > 0$ applied consistently to observed and simulated proxies.

### A.4    Optuna configuration (reproducibility)

All experiments use Optuna with the TPE sampler. For reproducibility we fix (i) the sampler seed, (ii) the number of trials $N_{\text{trials}}$, and (iii) the bounded search domain (Section 7), and we report the best trial parameters $\theta^\star$ together with fit metrics.

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
