# OpenReview forum: "Optimal Control of the Solow Model with an Epidemic Factor Calibrated to Real Data"
_mathai.club/MathAI/2026/Conference — 2026 Oral_

### Official Review · Reviewer_BxjA · 2026-03-12
**Calibration and Identifiability of a Coupled Solow–SIR Economic–Epidemiological Model under Sparse Annual Observations**

**Rating:** 6
**Confidence:** 4

**Review:**

Summary:
This paper proposes a coupled economic–epidemiological modeling framework combining a classical SIR epidemic model with a Solow-type economic growth equation. The work focuses on calibrating such models when only sparse annual observations are available, a challenging and practically relevant inverse problem. The authors develop a reproducible pipeline consisting of a forward simulator with within-year sub-stepping and an inverse calibration procedure based on a profiled likelihood objective combined with temporal regularization.

Strengths:
- The paper addresses an important interdisciplinary topic linking epidemic dynamics and economic modeling.
- The coupled Solow–SIR formulation provides a transparent and interpretable mechanistic framework for studying the interaction between epidemic burden and economic output.
- The calibration methodology is carefully structured, including the observation model, the objective function based on profiled likelihood, and the use of regularization to improve stability under short time series.
- The numerical pipeline is clearly described and reproducible, including the use of Optuna for robust parameter search in nonconvex calibration problems.
- The paper provides a concrete regional case study demonstrating the feasibility of simultaneously fitting economic and epidemiological time series within a single mechanistic model.

Suggestions for improvement:
The paper could be further strengthened by:
- expanding the empirical evaluation to additional case studies or datasets;
- discussing identifiability properties in greater depth (possibly with diagnostic experiments);
- clarifying how the proposed framework could support future extensions such as optimal control or stochastic formulations.

Final Recommendation:
POSTED / Poster-style acceptance with minor revision

Overall, the paper presents a clear and well-structured modeling and calibration framework that may serve as a useful baseline for future research on coupled economic–epidemiological systems and their data-driven calibration under sparse observations.

---

### Official Review · Reviewer_pmWf · 2026-03-13
**The paper presents a clear and reproducible calibration framework for a coupled Solow–SIR economic–epidemiological model under sparse annual observations**

**Rating:** 6
**Confidence:** 4

**Review:**

The paper presents a coupled economic–epidemiological model combining an SIR epidemic block with a reduced-form Solow-type economic model. Its main contribution is a calibration pipeline designed for sparse annual observations, including forward simulation with within-year sub-stepping and inverse estimation via a profiled likelihood objective with regularization.

Strengths

The paper addresses a relevant interdisciplinary problem. The proposed pipeline is clearly structured and reproducible.

Weaknesses

The economic block is quite simplified, which limits the strength of economic interpretation.
Several parameters appear to be strongly correlated and only weakly identifiable under annual observations.
Most importantly, correlation should not be confused with causation: even if the model jointly fits economic and epidemiological trends, this does not establish a causal relationship between epidemic burden and economic outcomes.

Recommendation

Overall, this is a clear and methodologically solid baseline paper. Its main value is in providing a reproducible calibration framework under sparse observations.

---

### Decision · Program_Chairs · 2026-03-14

**Decision:**

Accept (Oral)

**Comment:**

Dear Author(s),

On behalf of the Program Committee of the International Conference on Mathematics of Artificial Intelligence (MathAI 2026), we are pleased to inform you that your paper has been accepted for an oral presentation at MathAI 2026.

Your paper was evaluated through a rigorous two-stage review process involving both automated screening and expert review by members of the Program Committee. The reviewers recognized the quality and contribution of your work.

Presentation details:

- Format: Oral presentation (15–20 minutes + 5 minutes Q&A)
- Mode: You may present either in person (offline) at the conference venue in Sirius, Russia, or remotely via Zoom. Please indicate your preferred mode when confirming your participation.
- Conference dates: Marh 30 - April 3, 2026
- Website: https://mathai.club

Next steps:

1. Please confirm your participation and presentation mode by replying to this email mathai.club@yandex.ru no later than March 15, 2026 18:00 Moscow time.
2. If you plan to attend in person, the organizing committee will provide accommodation details separately.
3. Please prepare your final camera-ready manuscript according to the formatting guidelines available at https://mathai.club and upload it to OpenReview by March 15, 2026 18:00 Moscow time.

Should you have any questions regarding the program, logistics, or your presentation slot, please do not hesitate to contact us.

We look forward to your contribution to MathAI 2026.

With kind regards,

MathAI 2026 Program Committee
International Conference on Mathematics of Artificial Intelligence
https://mathai.club
OpenReview: https://openreview.net/group?id=mathai.club/MathAI/2026/Conference
Telegram: https://t.me/MathAI_club
Email: mathai.club@yandex.ru